# Differential axonal trafficking of Neuropeptide Y-, LAMP1-, and RAB7-tagged organelles in vivo

Joris P Nassal, Fiona H Murphy, Ruud F Toonen, Matthijs Verhage*

Departments of Functional Genomics and Clinical Genetics, Center for Neurogenomics and Cognitive Research (CNCR), VU University Amsterdam and VU University Medical Center, Amsterdam, Netherlands

**Abstract** Different organelles traveling through neurons exhibit distinct properties in vitro, but this has not been investigated in the intact mammalian brain. We established simultaneous dual color two-photon microscopy to visualize the trafficking of Neuropeptide Y (NPY)-, LAMP1-, and RAB7-tagged organelles in thalamocortical axons imaged in mouse cortex in vivo. This revealed that LAMP1- and RAB7-tagged organelles move significantly faster than NPY-tagged organelles in both anterograde and retrograde direction. NPY traveled more selectively in anterograde direction than LAMP1 and RAB7. By using a synapse marker and a calcium sensor, we further investigated the transport dynamics of NPY-tagged organelles. We found that these organelles slow down and pause at synapses. In contrast to previous in vitro studies, a significant increase of transport speed was observed after spontaneous activity and elevated calcium levels in vivo as well as electrically stimulated activity in acute brain slices. Together, we show a remarkable diversity in speeds and properties of three axonal organelle marker in vivo that differ from properties previously observed in vitro.

## Editor's evaluation

This is an important, well-written and easily comprehended quantitative imaging study that analyzes the motion of endo-lysosomal compartments within axons in vivo using simultaneous multiphoton imaging in the mammalian brain. The simultaneous dual two-photon imaging is well-executed and represents a substantive advance in a field that relies heavily on in vitro neuronal culture preparations. The authors address an issue of cell polarity, providing strong support for their ability to determine directional movement (anterograde versus retrograde), and characterize interesting differences in motion, including activity-dependent and calcium-dependent alterations at or near synapses.

**\*For correspondence:**
matthijs@cncr.vu.nl

**Competing interest:** The authors declare that no competing interests exist.

## Introduction

Neurons are highly compartmentalized cells which can span over large distances. Intracellular transport provides compartments with a variety of cargo necessary for cell homeostasis, growth, and inter-neuron signaling. Intracellular transport requires cytoskeletal microtubules and molecular motors, which move organelles along the microtubules. Many motors for different cargo have been identified which produce a complex picture of diverse transport dynamics, described by a variety of previous in vitro studies (*Hirokawa et al., 2010*; *Goo et al., 2017*; *Macaskill et al., 2009*; *Silverman et al., 2005*; *Zhang et al., 2013*).

Neurons and more specifically axons contain a variety of mobile organelles which move bidirectionally along microtubules and include: (1) mitochondria, which provide energy in the form of ATP, control calcium homeostasis, and serve as a contact hub for many other organelles (*Brookes et al.,*

*2004*; *Schon and Przedborski, 2011*; *Wu et al., 2017*). Mitochondria are distributed from the soma into axons and dendrites. They undergo fission and fusion events, and their transport at synapses is activity dependent (*Chen and Chan, 2009*; *Macaskill et al., 2009*). (2) Dense core vesicles (DCVs) contain a variety of neuropeptides which have modulatory effects ranging from excitability changes of local circuits to shifts in behavior (*Bacci et al., 2002*; *Melzer et al., 2021*). They are produced at the cell soma, are delivered into the axons, and released mainly at synapses (*Wong et al., 2012*; *van de Bospoort et al., 2012*). We showed previously that axonal DCV transport in vivo is highly dynamic and that DCVs slow down in axonal boutons (*Knabbe et al., 2018*). (3) Lysosomes are acidic organelles primarily engaged in degradative processes, but also involved in different signaling pathways and cell homeostasis (*Ballabio and Bonifacino, 2020*). Lysosomal proteins are delivered into the axon by endo-lysosomal organelles. These can fuse with autophagosomes to form autolysosomes. Consequently, lysosomes mature and acidify while being transported back to the soma (*Ferguson, 2018*). (4) Endosome function involves the transport of activated receptors, recycling of signaling and sorting factors, and the control of membrane homeostasis (*Kuijpers et al., 2021*). The different endosome stages are dynamically transported throughout the axon.

Mitochondria are the most extensively studied organelles regarding transport patterns in vitro, where they exhibit highly dynamic transport patterns (*Misgeld and Schwarz, 2017*). However, recently it was shown that mitochondria in cortical axons in vivo are mainly stationary and are located close to synapses (*Silva et al., 2021*). This highlights the importance of studies in vivo. For the other axonal organelles described above, detailed motility data in adult functional circuits are largely missing.

In this study, we utilized dual color two-photon imaging to analyze the trafficking dynamics of three organelle markers: the DCV marker Neuropeptide Y (NPY), and the endo-lysosomal markers LAMP1 and RAB7 in thalamocortical axons in intact living brain. In addition, to gain insight into local DCV slow down at synapses, we combined DCV imaging with an axonal-targeted calcium sensor and a synapse marker. Our data reveal that the lysosomal/endosomal markers LAMP1 and RAB7 have higher velocities than the DCV vesicle marker NPY. We also show that transport of DCVs slows down in and close to synapses. Furthermore, we show that increased calcium levels lead to an increase of the transport speed of NPY-tagged organelles. Together, we provide an overview and comparison of trafficking characteristics of axonal organelles tagged with NPY, Lamp1, and RAB7 in vivo which differ from properties previously observed in vitro.

## Results

### Two-photon imaging of organelle trafficking in thalamocortical axons in vivo

To analyze the trafficking dynamics of three organelle markers in intact living brain, we expressed the fluorescently tagged organelle markers NPY, LAMP1, and RAB7 in the thalamus of adult mice via viral injections (*Figure 1A*). NPY tagged with fluorophores is efficiently sorted into DCVs in vitro and in vivo (*Ramamoorthy et al., 2011*; *Bharat et al., 2017*; *Gumy et al., 2017*; *Persoon et al., 2018*; *Knabbe et al., 2018*). LAMP1 is classically used as a lysosomal marker but is much more heterogeneously distributed over endo-lysosomal and autophagic compartments. These include endosomes, lysosomes, amphysomes, and autophagosomes (*Saftig and Klumperman, 2009*; *Cheng et al., 2018*). However, roughly 50% of mobile axonal LAMP1 positive organelles contain active cathepsin D and are therefore most likely lysosomes in vitro (*Farfel-Becker et al., 2019*). RAB7 is classically described as a late endosome marker but is similarly to LAMP1 also found in early endosomes, lysosomes, and multivesicular bodies (*Vanlandingham and Ceresa, 2009*; *Shearer and Petersen, 2019*).

Two-photon imaging of the tagged organelles in thalamocortical projecting axons in layer 1 of the cortex was performed through a chronic cranial window (*Figure 1A and B*). Time-lapse recordings identified axons that expressed mobile organelle markers (*Figure 1*). Kymographs of these axons revealed the complex trafficking of these organelles. Within a single axon, NPY-tagged organelles changed speed, overtook each other, passed by in different directions, changed direction, or paused (*Figure 1F and G*). Kymograph tracings were used to quantify trafficking characteristics such as speed, direction of transport, and absolute flux of tagged organelles (*Figure 1H1*). The transport direction of NPY-tagged organelles with the highest average velocity was considered the anterograde direction based on a previous study (*Knabbe et al., 2018*) and verified with the microtubule plus end

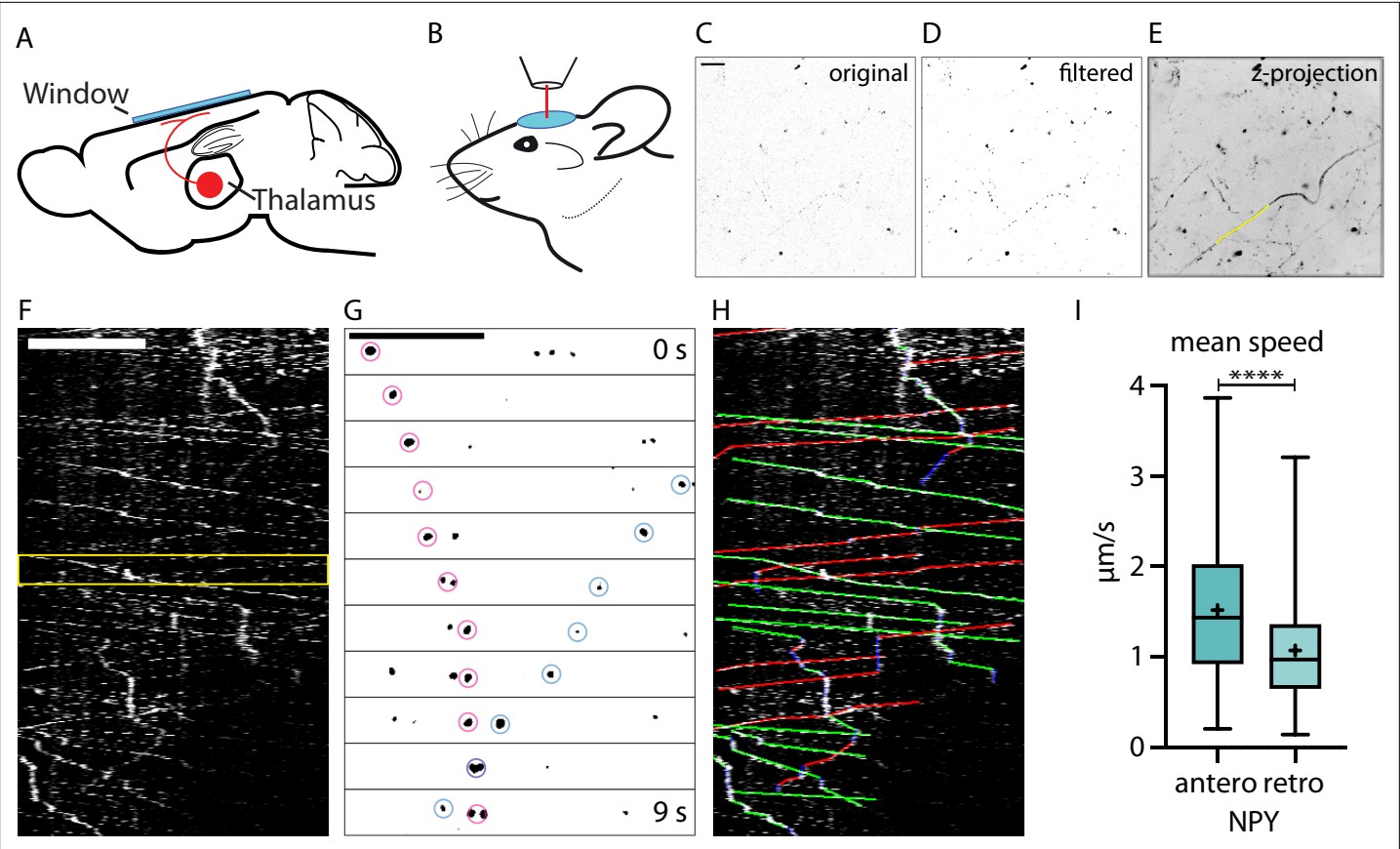

**Figure 1.** In vivo two-photon imaging of organelle dynamics in thalamocortical axons. (**A**) Adeno-associated virus (AAV) injections into the thalamus lead to the expression of fluorescent organelle markers in thalamocortical projections.(**B**) Implantation of a chronic cranial window enables two-photon imaging of thalamocortical axons in cortical layers 1 and 2. Typically, mice were imaged in 1 hr sessions starting at 3 weeks after injection. (**C**) Example raw data image of axons in cortical layer 1 expressing Neuropeptide Y (NPY)-Venus. Scalebar 20 μm. (D) Background subtracted and mean filtered version of C. (**E**) Z-projection of dataset from C (480 images acquired at 0.87 frames per second) with a sample organelle track in yellow. (**F**) Kymograph of an in vivo time-lapse recording. Scalebar 20 μm. (**G**) Zoom-in of the episode between the yellow lines in F. Two NPY-tagged organelles moving in opposite directions marked with pink and blue circles. Scale bar 20 μm. (**H**) Tracking of moving organelles in F with the faster moving organelles indicated in green (anterograde), the slower in red (retrograde), and pausing in blue. (I) Mean speed of 394 tracked NPY-tagged organelles in 25 axons in 5 mice. Anterograde mean speed 1.52 μm/s; retrograde mean speed 1.07 μm/s. Asterisks indicate level of significance (significance tested with Kolmogorov-Smirnov test, p-value<0.0001 ****).

marker MacF18 (Figure 3). Together, this approach enables the analysis of organelle trafficking in single cortical axons in vivo.

## LAMP1- and NPY-tagged organelles traffic through axons with different properties

To compare trafficking between LAMP1- and NPY-tagged organelles simultaneously in the same axon, NPY-Venus and LAMP1-mScarlet were co-expressed in thalamic neurons (*Figure 2A and B*; *Video 1*). Only moving fluorescent puncta were analyzed to prevent the inclusion of released fluorophore, organelles from other axons, and auto-fluorescent background. Virtually no overlap was observed between moving NPY- and LAMP1-tagged organelles.

LAMP1-tagged organelles moved significantly faster than NPY-tagged organelles with 2.37 vs 1.51 μm/s in the anterograde direction and 1.48 vs 1.11 μm/s in the retrograde direction (*Figure 2C*). Both markers moved in both directions, but NPY-tagged organelles moved 61% more often antero-gradely compared to 54% for LAMP1-tagged organelles (*Figure 2D*). Overall, there were 0.052 moving NPY-tagged organelles and 0.035 LAMP1-tagged organelles per μm axon per minute (*Figure 2E*). Pausing time (NPY 0.127 and Lamp1 0.13) between the two markers was similar (*Figure 2F*).

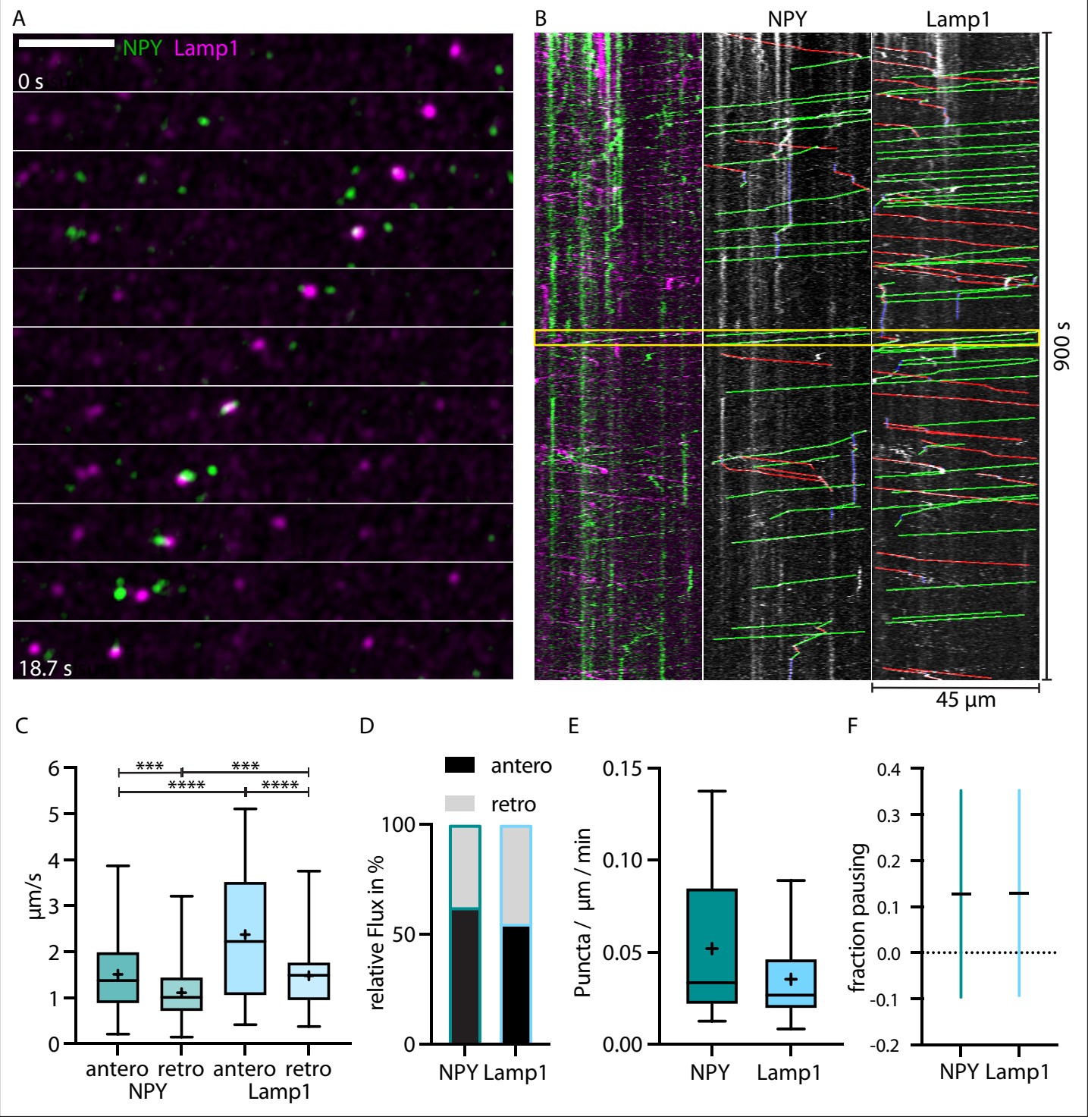

**Figure 2.** LAMP1-tagged organelles move faster than Neuropeptide Y (NPY)-tagged organelles in thalamocortical axons. (**A**) Zoom-in of in vivo time-lapse recording of an axon stretch in cortical layer 1 from thalamocortical neuron co-infected with NPY-Venus (green) and LAMP1-mScarlet (magenta) showing NPY- and LAMP1-tagged organelles passing each other. Scale bar 10 μm (**B**) Left: kymograph of the in vivo time-lapse recording depicted in A. The two yellow lines indicate the zoom-in area depicted in A. Middle: tracked moving NPY-tagged organelles in anterograde direction indicated in green, in retrograde direction in red, and pausing in blue. Right: tracked moving LAMP1-tagged organelles in anterograde direction indicated in green, in retrograde direction in red, and pausing in blue. (**C**) Mean speed of 159 NPY-tagged organelles and 122 LAMP1-tagged organelles in 7 axons in 4 mice. NPY mean speed: anterograde 1.51 μm/s and retrograde 1.11 μm/s. LAMP1 mean speed: anterograde 2.37 μm/s and retrograde 1.48 μm/s. Asterisks indicate level of significance (significance tested with Kolmogorov-Smirnov test, p-value<0.001 *** and <0.0001 ****). (**D**) Relative flux of the

*Figure 2 continued on next page*

*Figure 2 continued*

same NPY-and LAMP1-tagged organelles as in C. (**E**) Absolute flux (number of moving organelles per μm axon stretch per minute). Boxplots depicting distribution median (center line), mean (plus), and quartiles (top/bottom of box). (**F**) Mean fraction of pausing NPY- and LAMP1-tagged organelles. Is similar for NPY- and LAMP1-tagged organelles (0.127 for NPY and 0.13 for LAMP1). Error bars show SD.

Taken together, these data reveal a significant difference in the axonal trafficking properties between the DCV marker NPY and the endo-lysosomal marker LAMP1.

## Lamp1-tagged organelles move on average faster in the same direction as microtubules-plus-end marker MacF18

To confirm the proposed direction of the faster and slower Lamp1 trafficking directions, we co-expressed Lamp1-mScarlet with the microtubule plus-end marker MacF18-GFP (*Figure 3A*) in single thalamic neurons in vivo. MacF18 binds to the plus-end of growing microtubules which in axons are only directed in anterograde direction (*Figure 3C*). In every axon, the direction of the average faster trafficking Lamp1 puncta was the same as the anterograde (toward plus-end) direction of the MacF18-GFP puncta (*Figure 3B–D*). Hence, Lamp1-tagged organelles consistently move faster in anterograde than in retrograde direction, and the faster average direction can be considered the anterograde direction.

## RAB7- and NPY-tagged organelles move through axons with different properties

To compare trafficking between RAB7- and NPY-tagged organelles, RAB7-mScarlet and NPY-Venus were co-expressed in single thalamic neurons in vivo. (*Figure 4A and B*; *Video 2*). RAB7-marked organelles were both stationary and mobile. Mobile RAB7-tagged organelles were distinct from NPY-organelles as we barely detected any overlay between mobile RAB7-mScarlet and NPY-Venus. RAB7-tagged organelles moved significantly faster than NPY-tagged organelles with 2.34 vs 1.52 μm/s in the anterograde direction and 1.37 vs 1.02 μm/s in the retrograde direction (*Figure 4C*). Similar to LAMP1-tagged organelles, RAB7-tagged organelles moved equally often in both directions, while NPY-organelles moved more often in anterograde direction (*Figure 4D*).

Overall, 0.025 moving NPY positive puncta and 0.017 RAB7 positive puncta per μm axon per minute were detected (*Figure 4E*). In this experiment, NPY-tagged organelles paused more often than Rab7-tagged organelles (*Figure 4F*).

Together, these data reveal large differences in the axonal trafficking speed, the flux and pausing of the DCV marker NPY, and the endo-lysosomal marker RAB7.

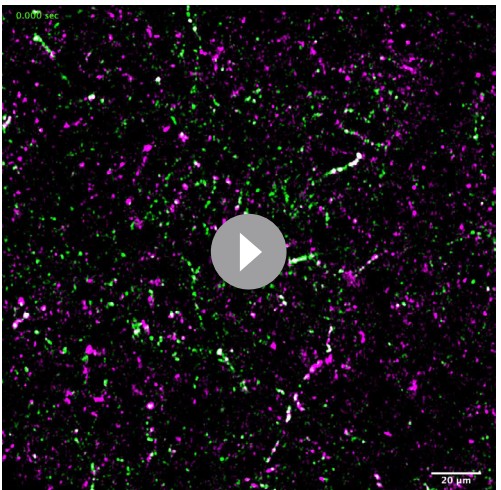

**Video 1.** Mobile Neuropeptide Y (NPY)- and LAMP1-tagged organelles in thalamic neurons.
https://elifesciences.org/articles/81721/figures#video1

## DCVs pause and slow down at synapses

We previously demonstrated that moving NPY-tagged organelles have the tendency to slow down at or near locations with morphological features of axonal varicosities (*Knabbe et al., 2018*). To identify these locations, we co-expressed NPY-Venus and Synapsin-mScarlet in single thalamic neurons in vivo. Synapsin labels presynaptic release sites (*Gitler et al., 2004*). Accordingly, the mScarlet signal was used to define synapses (*Figure 5A–C*). We compared trafficking speeds of NPY-tagged organelles within defined areas around synapses and the shaft (area outside the synapse mask) with different radii. NPY-tagged organelles paused (speed = 0) 50% more often than predicted by chance in the synapse area defined with a 0.5 μm radius but only 9% more often in a synapse area of 1.5 μm radius (*Figure 5D*). The effect was not

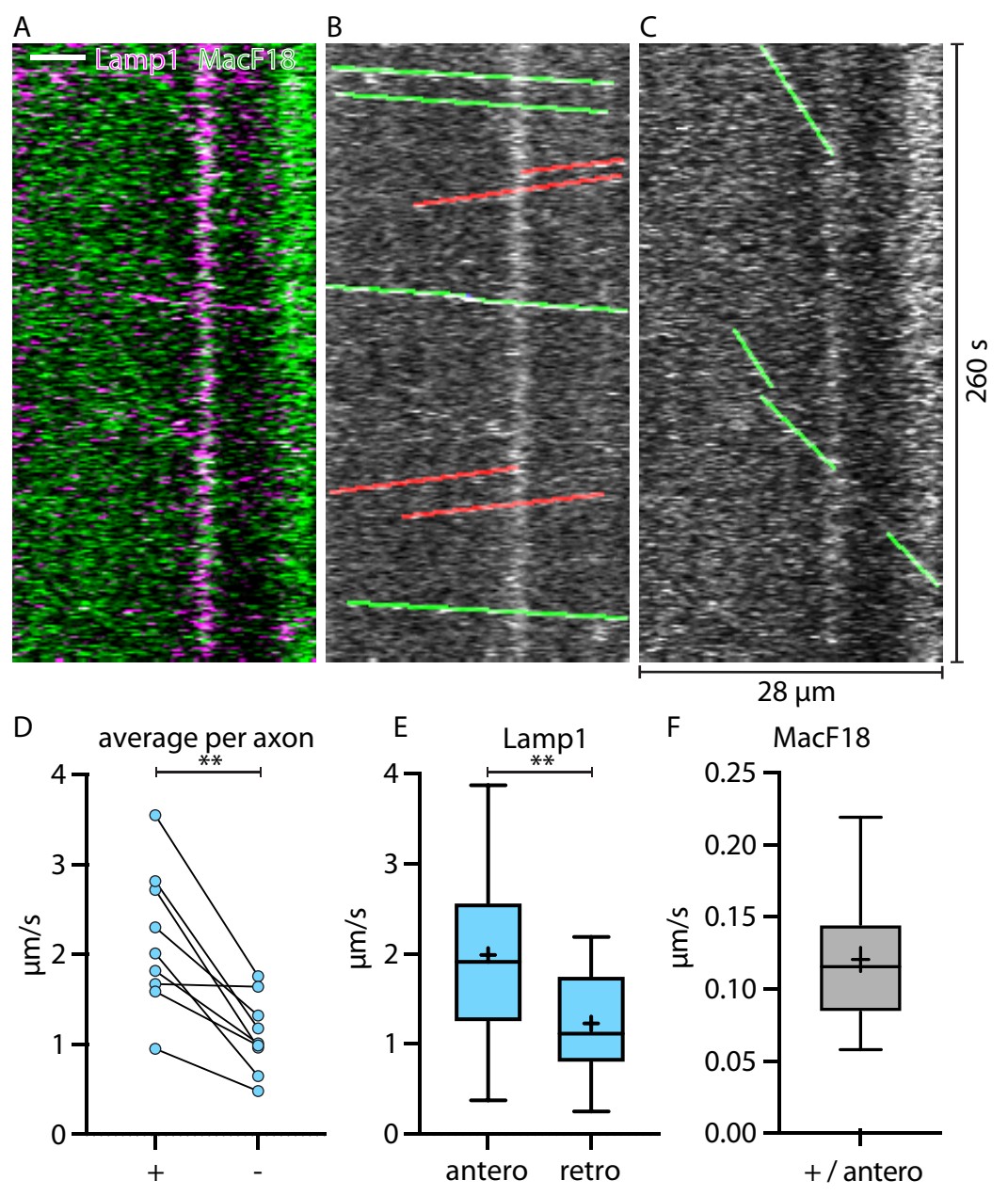

**Figure 3.** Lamp1-tagged organelles travel faster in anterograde direction, indicated with the microtubule plus-end marker MacF18. (**A**) Kymograph of an in vivo time-lapse recording of an axon stretch co-expressing Lamp1-mScarlet and MacF18-GFP. Scale bar 5 μm. (**B**) Lamp1 channel from A. Tracked Lamp1-tagged organelles with faster speed overlayed in green and slower overlayed in red. (**C**) MacF18 channel from A. Tracked MacF18 puncta overlayed in green indicating anterograde movement. (**D**) Paired analysis of average speeds in + and − end microtubule direction in individual axons (nine different axons) (**E**) Mean speed of Lamp1-tagged organelles in MacF18 co-expressing axons (9 different axons and 78 tracks). Asterisks indicate level of significance (significance tested with Wilcoxon signed-rank test, p-value<0.01 **). Asterisks indicate level of significance (significance tested with Kolmogorov-Smirnov test, p-value<0.01 **). (**F**) Mean speed of MacF18 puncta in with Lamp1 co-expressing axons (9 different axons and 36 tracks).

only detectable for organelles that stopped completely but also for organelles with slow moving speeds (*Figure 5E*). NPY-tagged organelles moved on average significantly slower in and close to synapses (radius 0.5 and 1.5 μm; *Figure 5F*). The analysis of anterograde and retrograde movement of NPY-tagged organelles within the same synapse masks showed that they slow down in and close to

**Figure 4.** RAB7-tagged organelles move faster than Neuropeptide Y (NPY)-tagged organelles in both directions. (**A**) Zoom-in of vivo time-lapse of axon co-expressing NPY-Venus (green) and RAB7-mScarlet (magenta) showing NPY- and RAB7-tagged organelles moving in opposite direction crossing each other. Scale bar 10 µm. (**B**) Left: kymograph of same in vivo time-lapse recording as the axon stretch depicted in A. Yellow box indicates zoom-in area depicted in A. Middle: tracked moving NPY-tagged organelles in anterograde direction indicated in green, retrograde direction in red, and pausing in blue. Right: tracked moving RAB7-tagged organelles in anterograde direction indicated in green, retrograde direction in red, and pausing in blue. (**C**) Mean speed of 244 NPY-tagged organelles and 195 RAB7-tagged organelles in 18 axons in 2 mice. NPY mean speed: anterograde 1.52 µm/s and retrograde 1.02 µm/s. LAMP1 mean speed: anterograde 2.34 µm/s and retrograde 1.37 µm/s. Asterisks indicate level of significance (significance tested with Kolmogorov-Smirnov test, p-value<0.05 *, <0.01 **, <0.001 ***, and <0.0001 ****). (**D**) Relative flux of the same NPY- and RAB7-tagged organelles as in C. (**E**) Absolute flux per µm axon stretch per minute. (**F**) Mean fraction of pausing NPY- and RAB7-tagged organelles. 0.22 for NPY and 0.085 for RAB7. Error bars show SD. Asterisks indicate level of significance (significance tested with Kolmogorov-Smirnov test, p-value<0.0001 ****).

synapses was only significant for retrograde moving organelles (*Figure 5G and H*). These data show that DCVs slow down and stop more often in and near synapses and that this slow down specifically affects retrogradely transported DCV.

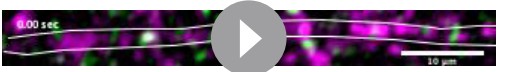

**Video 2.** Trafficking Neuropeptide Y (NPY)- and RAB7-tagged organelles in thalamic neurons.
https://elifesciences.org/articles/81721/figures#video2

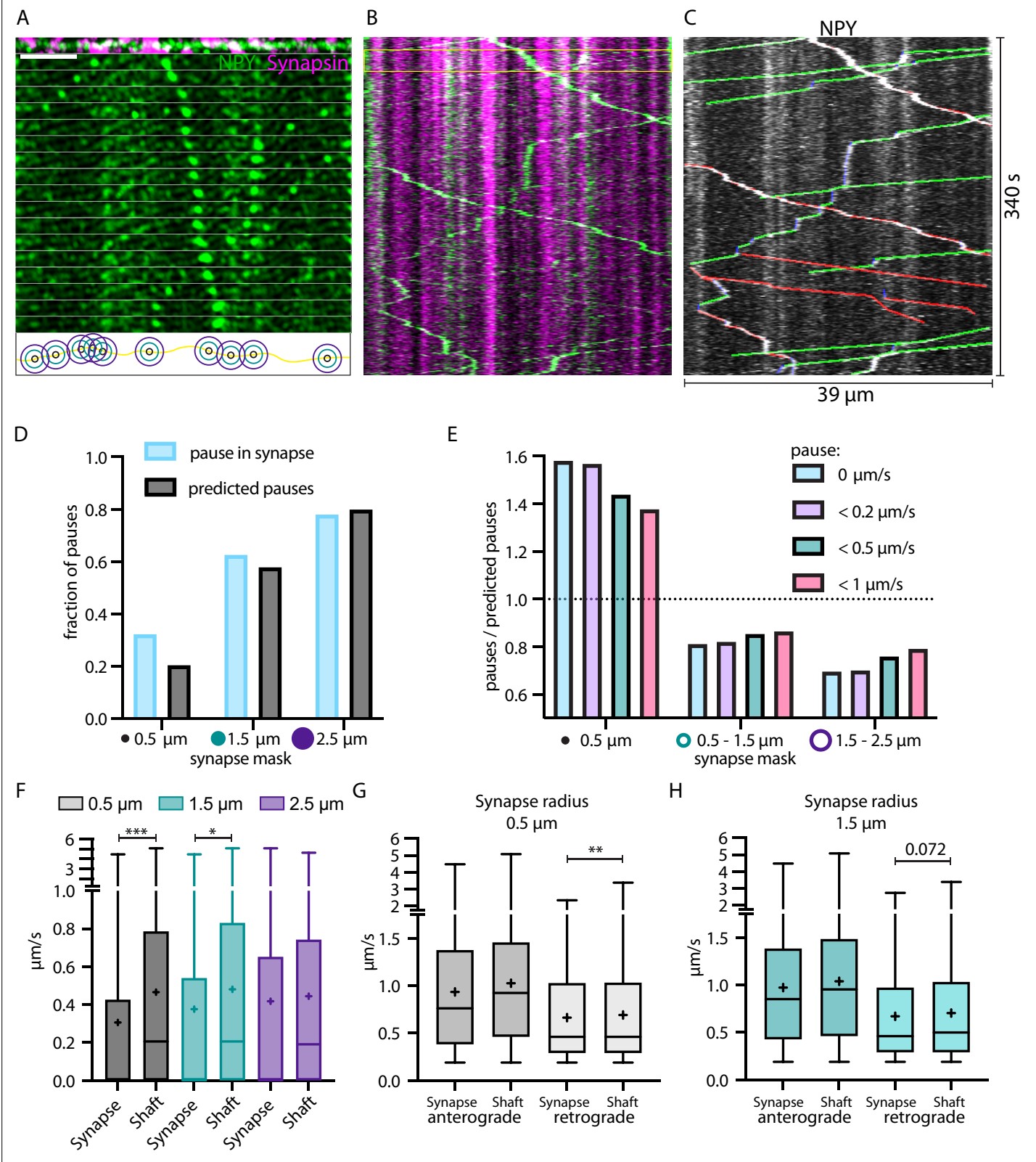

**Figure 5.** Neuropeptide Y (NPY)-tagged organelles slow down in Synapsin-labeled synapses. (**A**) Top: max-projection of axon stretch co-expressing NPY-Venus and Synapsin-mScarlet. Middle: time-lapse of NPY-Venus in the same axon stretch. Bottom: scheme of increasing radius of synapse mask from midpoint of Synapsin signal. Scale bar 10 μm. (**B**) Kymograph of axon stretch from A with area of time-lapse from A marked in yellow. (**C**) Tracked kymograph from B. With anterograde direction indicated in green, retrograde in red, and pausing in blue. (**D**) Fraction of pauses (speed value = 0; in

*Figure 5 continued on next page*

*Figure 5 continued*

blue) within synapse mask with increasing radius compared to predicted pauses (fraction of synapse mask area of the whole axon; in black). (**E**) Factor of pauses/predicted pauses for synapse area of 0.5 µm, 0.5–1.5 µm, and 1.5–2.5 µm (donut shape) with changing definition of pauses, only 0 µm/s, <0.2 µm/s, <0.5 µm/s, and <1 µm/s. Factor of one corresponds to same amount of measured and predicted pauses. (**F**) Individual local speed values in shaft (outside mask) vs synapse (inside mask) with increasing radius of synapse mask. 6324 individual speed values from 72 tracks (five axons and two mice). Means in µm/s: 0.5 µm synapse 0.31, shaft 0.47; 1.5 µm synapse 0.38, shaft 0.48; 2.5 µm synapse 0.42, shaft 0.44. Differences tested with linear mixed effect model including mouse, slice, and track level as random effects and location (synapse/shaft) as fixed effect. ANOVA was used to test model including location with model excluding it. Asterisks indicate level of significance test, p-value<0.05 ≙ * and p-value<0.001 ≙ ***. (**G and H**) Values from F for 0.5 µm and 1.5 µm synapse mask split up into anterograde and retrograde direction. Means in µm/s: (**G**) anterograde synapse 0.94, shaft 1.03; retrograde synapse 0.66, shaft 0.69; (**H**) anterograde synapse 0.97, shaft 1.04; retrograde synapse 0.67, shaft 0.71. Differences tested with linear mixed effect model including mouse, slice, and track level as random effects and location (synapse/shaft) as fixed effect. ANOVA was used to test model including location to model excluding it. Asterisks indicate level of significance test, p-value<0.01 ≙ **.

## Traveling speed of NPY-tagged organelles is increased after axonal Ca²⁺ activity

In in vitro systems, neuronal activity and elevated intracellular Ca$^{2+}$ trigger trafficking arrest of several mobile organelles, including DCVs (*de Wit et al., 2006*; *Stucchi et al., 2018*; *Goo et al., 2017*; *Macaskill et al., 2009*). To test this in vivo, we co-expressed axonal-targeted GCaMP6 (axoGCaMP, *Figure 6A*) with a signaling-dead NPY (NPYsd; *Figure 6—figure supplement 1A*) variant tagged with mScarlet in single thalamic neurons in vivo (*Video 3*). The NPYsd variant was used to ensure that the released NPY had no effect on local circuits. In cultured hippocampal neurons, overexpressed NPY and NPYsd both co-localized to endogenous DCV markers chromogranin A and B to the same degree, indicating that the mutations used to induce signaling-dead NPY do not affect its packaging or localization to DCVs. (*Figure 6—figure supplement 1B–D*). AxoGCaMP was used to detect spontaneous calcium activity. An active state of an axoGCaMP-expressing axon was defined as time in which the axoGCaMP signal was above half-maximal ΔF/F (*Figure 6B*). The speed of NPYsd-tagged organelles was correlated to active/rest state in the same axon (*Figure 6—figure supplement 2A and B*). The axonal activity mainly occurred in bouts alternated with longer stretches without activity (10% of the time spent active; *Figure 6C*).

No significant difference was observed in the speed measured immediately before and at the beginning of a bout of activity (*Figure 6—figure supplement 2B–D*) and when comparing all speed values during the active and rest state (*Figure 6D*).

When separately analyzing anterograde and retrograde transported organelles, opposing trends were observed. Anterograde organelles followed the trend for slower during rest and faster during the active state of the axon, while retrograde organelles had the opposite trend. To test for possible delayed effects caused by elevated calcium, additional time windows were defined including 5 s and 30 s after each activity ended and added the speed values in those windows to the active state. In the time window +5 s after activity, no significant difference in speed was observed and the trends remained (*Figure 6E*). When adding 30 s to the active state window, the lower speed during rest was significant and the trend in retrograde transported NPY-tagged organelles reversed (*Figure 6F*). Also, the frequency distribution of organelle speeds during activity vs rest showed a complex distribution with only a small shift between active and rest and different activity windows (*Figure 6—figure supplement 2E–M*). These data suggest a complex relationship between intracellular calcium levels and trafficking of NPY-tagged organelles with a significantly lower speed during a prolonged phase (30 s) after activity.

## Traveling speed of NPY-tagged organelles is altered by electrical stimulation in brain slices

To further test the relationship between activity-dependent Ca$^{2+}$-increases and organelle mobility, we switched to mossy fiber axons in acute hippocampal slices. AxoGCamp and NPYsd-mScarlet were expressed in the dentate gyrus (DG). Mossy fiber bundles were imaged in the CA3 region roughly 500 µm away from the injection site where DG axons run on the (dark) background of uninfected CA3 neurons (*Video 4*). The axons were recorded before, during, and after a 10 Hz electrical stimulation applied with an electrode positioned at the DG (*Figure 7A–C*). Recordings with visible axoGCaMP signal during stimulation (multiple boutons >1 ΔF/F during stimulation) in the majority of axon

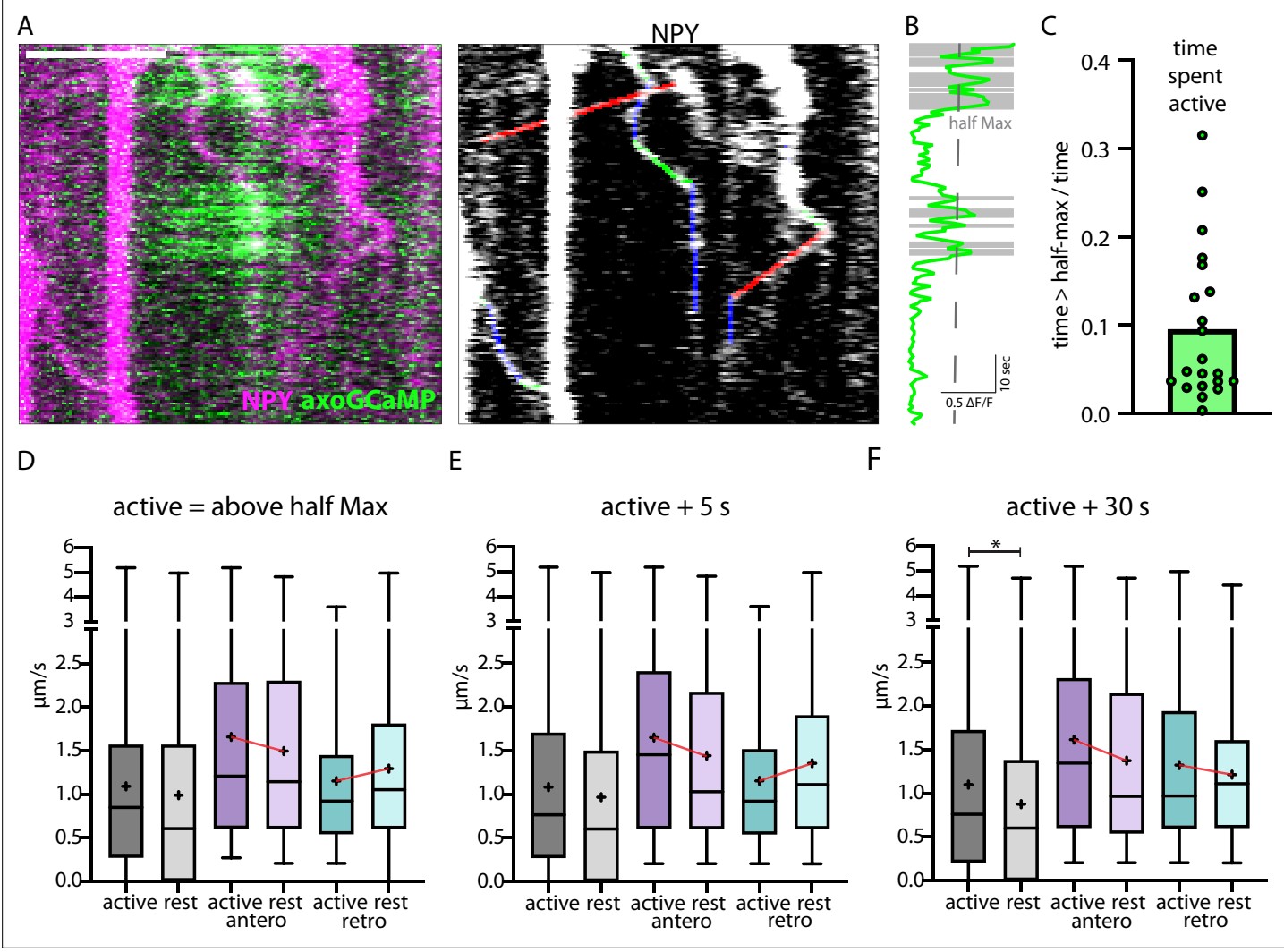

**Figure 6.** Correlation of axonal calcium influx measured with AxoGCamp and trafficking of Neuropeptide Y (NPY)-tagged organelles. (**A**) Kymograph of axon stretch co-expressing axoGCaMP (green) and NPY-mScarlet (magenta). Scale bar 10 µm. Right: tracked moving NPY-tagged organelles in one direction indicated in green, the other in red, and pausing in blue. (**B**) Calcium trace corresponding to A in green and above half maximal ΔF/F in gray. (**C**) Mean ratio of time the calcium level was above half max (21 axon stretches). The intensity of the maximal axoGCamp in between axon stretches varied from 0.2 to 1 ΔF/F. (**D**) Box plots of speed values (4313 speed values of 129 tracks in 18 axons from 8 datasets in 3 mice) of NPY-tagged organelles during activity and at rest (above and below half max of ΔF/F GCamp signal). Means in µm/s: active 1.09, rest 0.99, active antero 1.66, rest antero 1.50, active retro 1.15, and rest retro 1.30. (**E and F**) +5 and +30 depict the same tracks but with longer defined activity windows after ΔF/F falls below half max (5 s or 30 s longer window). Means for E in µm/s: active 1.08, rest 0.97, active antero 1.65, rest antero 1.44, active retro 1.15, and rest retro 1.36. Means for F in µm/s: active 1.10, rest 0.88, active antero 1.62, rest antero 1.38, active retro 1.33, and rest retro 1.22. Differences tested with linear mixed effect model including mouse, dataset, and track level as random effects and location (synapse/shaft) as fixed effect. ANOVA was used to test model including activity with model excluding it. Asterisks indicate level of significance test, p-value<0.05 *.

The online version of this article includes the following figure supplement(s) for figure 6:

**Figure supplement 1.** Neuropeptide Y (NPY)-pHluorin and signaling-dead NPY (NPYsd)-pHluorin both co-localize to endogenous dense core vesicle (DCV) markers.

**Figure supplement 2.** Speed of Neuropeptide Y (NPY)-tagged organelles correlated to activity state.

bundles were selected for analysis. In resting brain slices, frequent intrinsic activity was observed. The average speed of NPY-tagged organelles in mossy fibers was 0.90 µm/s before, 1.02 µm/s during, and 0.90 µm/s after elevated calcium levels elicited by electrical stimulation (*Figure 7D*, *Figure 7— figure supplement 1*). The speed values after stimulation were significantly lower than during (p-value 0.0002), while the difference between before and during is not. The difference could be explained by

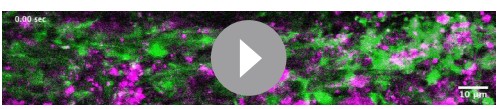

**Video 3.** Axonal-targeted GCamp6 and signaling-dead Neuropeptide Y (NPYsd)-mScarlet in a single thalamic neuron.
https://elifesciences.org/articles/81721/figures#video3

a higher variation before the stimulation due to higher and varying intrinsic activity. When separately analyzing the speed values for anterograde and retrograde transport, the speed values before and during stimulation were significantly different for anterograde transported NPY-tagged organelles. Together these data show that activity enhances DCV transport speed, especially for anterograde transport, in this ex vivo preparation.

## Discussion

The transport and location of different organelles in the cell are crucial for signaling, cell homeostasis, and growth. So far, organelle motility has mostly been studied in in vitro models. This study provides trafficking characteristics of NPY-, LAMP1-, and RAB7-tagged organelles in single thalamocortical axons in vivo. We show that NPY-tagged secretory organelles are transported significantly slower than LAMP1- and RAB7-tagged endo-lysosomal organelles in both directions (*Figure 8*). NPY-tagged organelles are transported more selectively in anterograde direction, while endo-lysosomal organelles travel in both directions equally. NPY-tagged organelles slow down and pause at axonal locations that accumulate Synapsin-mScarlet. Significant change of their transport speed was observed during spontaneous activity and elevated calcium levels in vivo as well as during electrically stimulated activity in acute brain slices.

Current theory of organelle trafficking relies almost exclusively on the overexpression of protein markers to tag different classes of organelles. However, overexpression of such markers might influence organelle identity, abundance, and localization. Our in vivo approach and thalamic injection site promote the selective analysis of those organelles that travel into the distal axon. Furthermore, the organelle marker proteins used in this study are among the best validated tools. NPY-GFP/pHluorin was previously shown to be sorted exclusively into DCVs, also in vivo (*Persoon et al., 2018*), to travel with the characteristics of DCVs in vitro, and to be in low pH compartments of DCV size. LAMP1-GFP labels a multitude of endo-lysosomal organelles. Recent studies report that most axonal LAMP1-tagged organelles are not acidic enough to be classified as lysosomes. These studies concluded that mature lysosomes do not typically enter the axon (*Lie et al., 2021*), while other studies demonstrated transport of degradative and LAMP1-positive lysosomes into axons (*Farfel-Becker et al., 2019*). Hence, it is still unclear whether degradative cargo is delivered by mature lysosomes or by other organelles throughout the axon, especially in vivo. Therefore, only some of the LAMP1-tagged organelles observed in the current study may be degradative lysosomes. RAB7 and LAMP1 might partially overlap in the organelles they label. However, the difference in some trafficking properties observed here suggests these markers label at least partially different subpopulations of organelles. Furthermore, no overlap between overexpressed NPY and LAMP1 as well as NPY and RAB7 was found in moving organelles, indicating that these markers label distinct classes of organelles: DCVs and two classes of endo-lysosomal organelles.

Previous studies have quantified organelle transport velocities in vitro with the same or similar markers this study utilizes. For DCVs labeled with NPY or other neuropeptides (BDNF (brain-derived neurotrophic factor), tPA, Sem3A, Ilp-2, and ANF (atrial naturietic factor)), anterograde velocities range from 0.78 µm/s, at 32 °C to 1.23 µm/s at 37° to 1.37 µm/s and from 0.43 µm/s to 1.42 µm/s in retrograde direction (*Barkus et al., 2008*; *Bittins et al., 2010*; *Cavolo et al., 2015*; *de Wit et al., 2006*; *Kwinter et al., 2009*; *Lund et al., 2021*; *Sung and Lloyd, 2022*, *Table 1*). The velocities shown here are slightly faster at least in anterograde direction, in line with what we previously reported in vivo (*Knabbe et al., 2018*). For axonal organelles labeled with LAMP1 in vitro, values from 0.9 µm/s to 2 µm/s in anterograde and from 0.8 µm/s to 1.5 µm/s in retrograde direction were reported (*Boecker et al., 2020*; *De Pace et al., 2020*; *Snouwaert et al., 2018*, *Table 1*). The LAMP1 velocities reported here are similar to the highest

**Video 4.** Mossy fiber bundles with axonal targeted GCamp6 and signaling-dead Neuropeptide Y (NPYsd)-mScarlet in acute hippocampal slices.
https://elifesciences.org/articles/81721/figures#video4

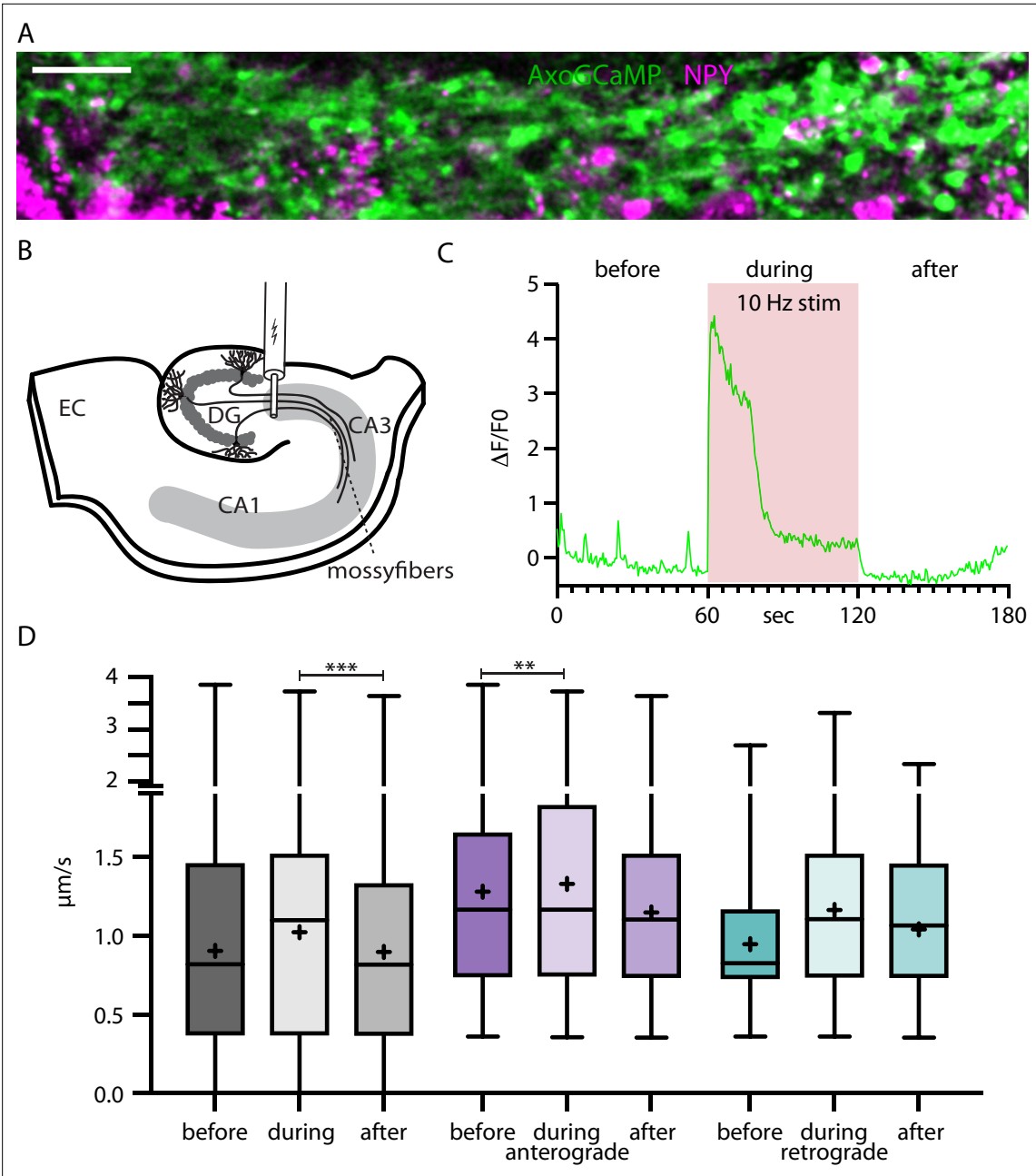

**Figure 7.** Neuropeptide Y (NPY)-tagged organelle trafficking during electrical stimulation in acute hippocampal slices. (**A**) Max-projection of dentate gyrus (DG) mossy fibers in the CA3 region of the hippocampus co-expressing axoGCaMP and NPY-mScarlet in an acute-slice. Scale bar 20 μm. (**B**) Scheme of acute slice electrical stimulation in DG/CA3 region. Adeno-associated virus (AAV) axoGCaMP and NPY-mScarlet injected in DG. (**C**) Example axoGCaMP trace of a mossy-fiber bouton with 1 min 10 Hz stimulation (red box). (**D**) Box plots of speed values (5446 speed values of 247 tracks in 100 axons from 5 slices from 2 mice) tracked NPY-tagged organelles divided into before, during, and after the 1 min long 10 Hz electrical stimulation. Gray: pooled values; violet: speed values from anterograde transported organelles; green: speed values from retrograde transported organelles. Differences tested with linear mixed effect model including mouse, dataset, and track level as random effects and location (synapse/shaft) as fixed effect. ANOVA was used to test model including activity with model excluding it. Asterisks indicate level of significance test, p-value<0.01 **, and p-value<0.001 ***.

The online version of this article includes the following figure supplement(s) for figure 7:

**Figure supplement 1.** Speed distribution of Neuropeptide Y (NPY)-tagged organelles across activity states.

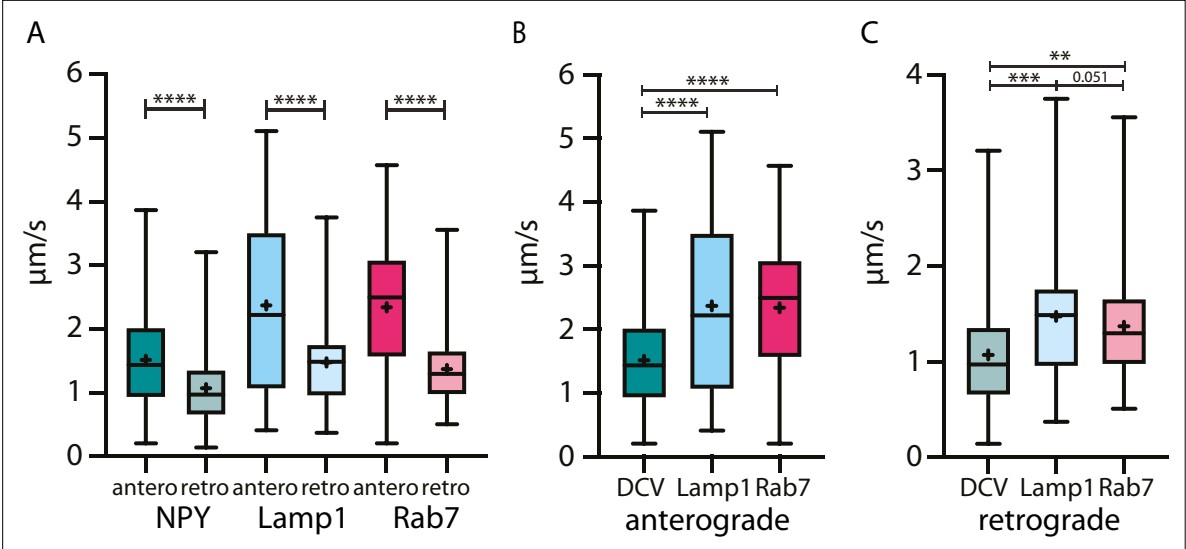

**Figure 8.** The velocity of three different organelle markers in vivo speed from Neuropeptide Y (NPY)-, LAMP1-, and RAB7-tagged organelles in anterograde and retrograde direction. (**A**) All markers traveled significantly faster in anterograde than in retrograde direction. (**B and C**): NPY-tagged organelles moved significantly slower than both endo-lysosomal markers in both directions. Asterisks indicate level of significance (significance tested with Kolmogorov-Smirnov test, p-value<0.05 *, <0.01 **, <0.001 ***, and <0.0001 ****).

of the reported in vitro velocities. Reports about fluorescently tagged axonal RAB7 velocity in cultured cells and non-mammalian model systems range from 0.31 µm/s to 1 µm/s in anterograde and retrograde direction (*Boecker et al., 2020*; *Castle et al., 2014*; *Lund et al., 2021*; *Zhang et al., 2013*, *Table 1*). We observed considerably higher velocities for RAB7 in vivo as well as a distinct difference in anterograde and retrograde transport speeds. Notably, signaling endosomes travel in retrograde direction in the mouse sciatic nerve in vivo significantly faster than our reported values (*Sleigh et al., 2020*). Hence, while trafficking speeds for DCVs and LAMP1-tagged endo-lysosomal organelles are consistent between studies in different model systems, substantial differences exist for RAB7-tagged endo-lysosomal organelles.

All three classes of organelles here are reported to be transported anterogradely by, amongst others, KIF-1, the fastest of the anterograde motors (*Stucchi et al., 2018*; *Hummel and Hoogenraad, 2021*; *Bentley et al., 2015*). Retrograde transport is supported by the motor dynein (*Tan et al., 2011*; *Wong et al., 2012*). DCVs and the endo-lysosomal organelles showed substantial differences in trafficking speeds. Microtubule-based transport can be modulated in different ways, which might provide an explanation for the observed speed differences: On the levels of the motors (activation, inactivation, combination, and regulation) and each organelle has unique proteins coating its surface, meaning that interactions with adapter proteins and/or motor proteins may have different affinities and dynamics. Together, this could explain the complex trafficking characteristics described here.

We provide evidence that DCVs slow down at synapses, especially for retrograde transported organelles. Previous studies in rat neurons in vitro showed that synaptic microtubule organization and a lower affinity of KIF1A to microtubule plus-ends lead to the delivery of anterograde transported cargo to the synapse (*Guedes-Dias et al., 2019*). Other studies in fruit fly neuromuscular junction report low, inefficient bidirectional capture of DCV in all but the most distal boutons (*Wong et al., 2012*). The differences to our in vivo data could be explained by differences in model systems as well as the integration of multiple mechanisms on different trafficking directions.

In this study, we provide evidence for delayed, activity-dependent slow down of DCVs. The changing trend of retrograde transported DCVs from slowing down during immediate activity to speeding up in the 30 s after activity could explain the overall significant increase of velocity during activity. This would mean that there is a relatively slow mechanism which gets activated by activity affecting retrograde transported DCVs. The slice experiments in *Figure 7* support the argument of a different effect of activity on retrograde and anterograde DCV transport. Previously described activity-dependent changes in DCV motility in vitro have been explained by different pathways.

**Table 1.** Comparison of organelle marker trafficking characteristics in different model systems from literature.

| Model | Marker | Source | Speed in µm/s | | Fraction | Rel. flux in | Abs. flux in |
|---|---|---|---|---|---|---|---|
| | | | Antero | Retro | Pausing | Antero dir. | Org./µm/min |
| Mouse thalamocortical neurons | Neuropeptide Y (NPY) | Nassal et al. | 1.518 | 1.073 | 0.13/0.22 | 0.61/0.82 | 0.052/0.025 |
| Mouse thalamocortical neurons | NPY | Knabbe et al. | 2.14 | 1.4 | 0.11 | | |
| Mouse sciatic nerve motor neurons | HcT/Neurotrophin | Sleigh et al. | | 2.5–2.8* | 0.01* | | |
| Mouse sciatic nerve sensory neurons | HcT/Neurotrophin | Sleigh et al. | | 1.8* | 0.005* | | |
| Mouse thalamocortical neurons | LAMP1 | Nassal et al. | 2.371 | 1.476 | 0.1298 | 0.54 | 0.035 |
| Mouse thalamocortical neurons | RAB7 | Nassal et al. | 2.344 | 1.37 | 0.085 | 0.51 | 0.017 |
| Mouse cultured cortical neurons | Sema3A | de Wit et al. | 0.78 | 0.43 | 0.22 | | |
| Mouse cultured hippocampal neurons | NPY | Bittins et al. | 1.14 | 0.75 | | | 0.023* |
| Rat cultured hippocampal neurons | tPA | Kwinter et al. | 1.23 | 1.28 | 0.16 | | |
| Rat cultured hippocampal neurons | BDNF | Cavolo et al. | 1.37 | 1.42 | | | |
| Rat cultured hippocampal neurons | LAMP1 | Boecker et al. | 2 | 1.5* | | 0.5$§ | |
| Human iPSC derived cultured neurons | LAMP1 | Boecker et al. | 1.5 | 1* | | 0.5$§ | |
| Mouse cultured hippocampal neurons | LAMP1 | De Pace et al. | 1.7 | | | | |
| Mouse cultured hippocampal neurons | LAMP1 | Snouwaert et al. | 0.85* | 0.75* | | | |
| Rat cultured hippocampal neurons | RAB7 | Boecker et al. | 1* | 1* | | 0.29$§ | |
| Human iPSC (induced pluripotent stem cell) derived cultured neurons | RAB7 | Boecker et al. | 0.7* | 0.9* | | 0.25$§ | |
| Rat cultured cortical neurons | RAB7 | Castle et al. | 0.57 | 0.57 | 0.73/0.62§ | | |
| Rat cultured DRG neurons | RAB7 | Zhang et al. | 0.31 | 0.53 | | 0.33$ | |
| *Drosophila* larval motor neurons | Ilp-2 | Lund et al. | 0.9* | 0.75* | 0.1* | 0.4* | |
| *Drosophila* larval motor neurons | ANF | Sung et al. | 0.78* | 0.78* | | | |
| *Drosophila* larval motor neurons | RAB7 | Lund et al. | 0.9* | 0.85* | 0.15* | 0.25* | |
| *Drosophila* larval motor neurons | Spinster | Lund et al. | 0.95* | 0.85* | 0.1* | 0.4* | |
| *C. elegans* larval motor neurons | ANF | Barkus et al. | 1.14 | 0.41 | 0.13 | | |

*Values have been estimated from graph in figures, §comparability issues, $values were transformed from original for better comparability.

One proposed mechanism is the activity-dependent phosphorylation of JNK (c-Jun N-terminal kinase) which in turn phosphorylates Synaptotagmin-4 and thereby destabilizes the interaction with KIF1 leading to DCV capture at synapses (*Bharat et al., 2017*). A different activity-dependent recruitment for DCV at synapses was proposed in dendrites. The KIF1 binding partner Calmodulin senses calcium and leads to binding and increased mobility of DCVs (*Stucchi et al., 2018*). Multiple proposed mechanisms for organelle speed increase and decrease could operate in parallel and/ or act on different subpopulations of DCVs (*Cason and Holzbaur, 2022*). This could obscure a more significant detection of a faster or more direct effect of activity on DCV transport. Recently, activity-dependent slow down was shown for mitochondria in vivo. The study also reported in cultured slices that the arrest of mitochondria in excitatory neurons is not fully dependent on glutamate-activated calcium influx, as previously proposed. Therefore, another unknown factor co-released with neurotransmitter was proposed (*Silva et al., 2021*). This could also be the case for DCVs.

## Materials and methods

**Key resources table**

| Reagent type (species) or resource | Designation | Source or reference | Identifiers | Additional information |
|---|---|---|---|---|
| Genetic reagent (*M. musculus*) | C57Bl/6 J | Charles River | C57Bl/6 J Strain code:632 | |
| Antibody | MAP2 (chicken, monoclonal) | Abcam | ab5392 | 1:500 |
| Antibody | Chromogranin B (rabbit, polyclonal) | Synaptic Systems | SySy 259103 | 1:500 |
| Antibody | Chromogranin A (rabbit, polyclonal) | Synaptic Systems | SySy 259003 | 1:500 |
| Software and algorithm | ImageJ/Fiji | ImageJ | RRID:SCR 002285 | - |
| Software and algorithm | Prism | Graphpad | RRID:SCR 002798 | - |
| Software and algorithm | R/RStudio | The R Foundation | RRID:SCR_0019057 RRID:SCR_000432 | - |

## Animals

All animal experiments were approved by the local animal research committee of the VU University, Amsterdam and were carried out in accordance with the European Communities Council Directive (010/63/EU). Experiments were conducted with wildtype (WT) C57Bl/6 N mice of both sexes with an age range from 8 to 12 weeks at the beginning of the experiments. The mice were kept at a 12–12 hr dark–light cycle. Water and food were available ad libitum, except during imaging sessions. Mice were generally housed in groups up to three animals per cage but were separated after surgery to prevent injuries. After completion of the experiments, mice were deeply anesthetized with avertin and transcardially perfused or decapitated for slice experiments.

## Fluorescent labeling by overexpression of fusion proteins

### In vitro

NPY-pHluorin was made by replacement of NPY-Venus with pHluorin. NPY(sd)-pHluorin was generated by removing the CPON sequence in NPY and mutating the last two amino acids that constitute the GKR receptor binding site. The final construct sequence is: MLGNKRLGLSGLTLALSLLVCLGALAEA YPSKPDNPGEDAPAEDMARYYSALRHYINLITRQAA. All constructs were driven by a synapsin promoter, sequence verified, cloned into a pLenti vector, and produced as described previously (*Naldini et al., 1996*).

### In vivo

List of viral vectors with constructs:

> NPY:
> AAV2; hSynapsin1(promotor)-NPY-Venus
> AAV5; hSynapsin1(promotor)-NPY(signaldead)-mScarlet1
> LAMP1:
> AAV5; hSynapsin1(promotor)-LAMP1-mScarlet1
> RAB7:
> AAV5; hSynapsin1(promotor)-mScarlet1-RAB7
> Synapsin:
> AAV5; hSynapsin1(promotor)-Synapsin1-mScarlet1
> MacF18:
> AAV5; hSynapsin1(promotor)-EGFP-MacF18
> GCaMP:
> AAV1; hSynapsin1(promotor)-axon-GCaMP6s

Organelles were labeled in vivo in mice by viral expression of a fusion protein consisting of NPY, Lysosomal-associated membrane protein 1 (LAMP1), Ras-related protein 7 (RAB7), and the fluorescent proteins Venus (e.g. NPY-Venus) or mScarlet1 (e.g. NPYsd-mScarlet; *Figure 1A and B*) driven by the synapsin promoter. Microtubule plus (+)-ends were marked by expression of green fluorescent protein (EGFP) fused to the first 18 N-terminal amino acid residues of the microtubule +-end marking protein

MACF43 fused to the two-stranded leucine zipper coiled-coil sequence corresponding to GCN4-p1 under control of the synapsin promoter as described elsewhere (*Yau et al., 2016*; MACF18-GFP). Synapses were labeled by expression of Synapsin1 fused with mScarlet1 under a Synapsin promotor. To visualize calcium levels, we expressed a fusion protein of the GAP43 palmitoylation site (axonal target signal) and the calcium sensor GCaMP6s. This was a gift from Lin Tian (Addgene viral prep # 111262-AAV1; http://n2t.net/addgene:111262; RRID:Addgene_111262). All construct were packed into recombinant adeno-associated virus (AAV) particles of serotype 2, 5, and 1 (see list).

## Stereotaxic injection and chronic window implantation for in vivo imaging

For viral injection and craniectomy (protocol adapted from *Holtmaat et al., 2009*), mice were anesthetized with an intraperitoneal (I.P.) injection of a mixture of fentanyl (Fentadon; Dechra, Northwhich, United Kingdom) 0.05 mg/kg, midazolam (Midazolam; Actavis, Dublin, Ireland) 5 mg/kg, and medetomidine (Sedastart; ASTfarma, Oudewater, Netherlands) 0.5 mg/kg body weight each diluted in saline and placed in a stereotaxic head holder (Kopf, Tujunga, CA, USA). To prevent inflammation, animals were given dexamethasone (Dexdomitor; Vetoquinol B.V., Breda, Netherlands) 2 mg/kg intramuscular. Depth of anesthesia was monitored by checking breathing frequency and pain reflexes. Before cutting of the skin, lidocaine (Lidocaine; Dechra, Northwhich, United Kingdom) solution (1%) was injected subcutaneously. A small hole in the skull was made with a dental drill (0.5 mm ball drill bit; Meissinger, Neuss, Germany) over the injection site. Then, via microinjection glass needles, 1 µl of a 1:1 mixture of two different AAV particles (titer was adjusted beforehand to inject a similar amount of AAV particles) was slowly injected into the right medio-dorsal thalamus (coordinates from bregma: $x$=1.13, $y$=–0.82, and $z$=–3.28). Afterward, a circular craniectomy (approx. 6 mm, center positioned 1 mm right of bregma) was drawn with a dental drill, and a circular skull segment was removed. Sterile Ringer solution was used to keep brain surface wet. The dura was carefully removed. A sterile round no. 0 coverslip with a diameter of 6 mm (cranial window) and a custom-made round plastic holder surrounding it for head fixation were cemented on the skull with a mixture of dental cement powder (Paladur; Kulzer, Hanau, Germany) and superglue (Pattex; Düsseldorf, Germany).

After surgery, mice received I.P. a mixture of naloxone (Naloxon HCI-hameln; Hameln Pharma Plus GmbH, Hameln, Germany) 1.2 mg/kg, flumazenil (Flumazenil Kabi, Fresenius Kabi, Bad Homburg, Germany) 0.5 mg/kg, and atipamezole (Sedastop; ASTfarma) 2.5 mg/kg body weight diluted in saline (NaCl 0.9%, Fresenius Kabi) to antagonize the anesthesia. Mice were given carprofen (Rimadyl; Pfizer, New York City, New York) in the drinking water 1 day before surgery and during the next 3 days. Mice were single housed after surgery and typically imaged at least 21 days after the surgery. This waiting period is essential for the glial reaction below the window to subside (*Holtmaat et al., 2009*).

## In vivo two-photon imaging

Two-photon imaging (*Denk et al., 1994*) was performed with a TriM Scope I microscope (LaVision BioTec GmbH, Bielefeld, Germany) equipped with a pulsed Ti:Sapphire laser (Chameleon; Coherent, Santa Clara, CA, USA). 960–980 nm excitation light was used to simultaneously excite the pairs of green and red fluorophores. Imaging was performed with a 25× water immersion objective (Nikon MRD77225, NA = 1.1; Minato, Japan) and appropriate filter sets (530/55, 650/100; Brightline Semrock, West Henrietta, NY, USA). Fluorescence emission was detected with low-noise high-sensitivity photomultiplier tubes (PMTs, H7422-A40; Hamamatsu Photonics K.K., Hamamatsu, Japan). To anesthetize the animals for imaging, anesthesia was first induced with 5% isoflurane in oxygen and maintained at 0.8–1%. Depth of anesthesia was monitored by checking for breathing frequency and pain reflexes. The anesthetized mouse was head fixed by clamping the plastic holder cemented to the skull. Frames were typically taken from an area of 196.6×196.6 µm at 1024×1024 pixel resolution with a frame rate of 1.06 Hz for 10 min per dataset using the galvanometric scanner of the TriM Scope I. Anesthetized imaging sessions lasted no longer than 1 hr.

## Stereotactic injection and acute slice preparation for slice imaging

Before surgery, mice received 0.1 mg/kg buprenorphine (Temgesic; RB Pharma, Lisboa, Portugal). Mice were anesthetized with isoflurane. Depth of anesthesia was monitored by checking breathing frequency and pain reflexes. Before cutting of the skin, lidocaine solution (1%) was injected subcutaneously. A

small hole in the skull was made with a dental drill over the injection site. To target hippocampal mossy-fibers, 50 nl AAV mixture was bilaterally injected into the DG of the 'lateral' hippocampus (coordinates from bregma: $x$=2.1, $y$=–3.1, and $z$=–2.3) via a microinjection glass needle. After the AAV injection, the skin was sutured close. Animals received carprofen (Rimadyl) in the drinking water 1 day before surgery and during the next 3 days.

After 3–6 weeks, animals were killed, and the brains were immediately transferred into ice-cold high sucrose slicing ACSF (artifical cerebralspinal fluid; 70 mM NaCl, 2.5 mM KCl, 1.25 mM $NaH_2PO_4 \cdot H_2O$, 5 mM $MgSO_4 \cdot 7H_2O$, 1 mM $CaCl_2$, 25 mM $NaHCO_3$, 70 mM sucrose, 25 mM glucose, 1 mM sodium ascorbate, and 3 mM sodium pyruvate). 350 µm thick slice horizontal slices were made on a Leica VT1200S vibratome. Slices are slightly angled to allow for completely intact mossy-fibers from DG to the end of CA3 (magic cut, *Bischofberger et al., 2006*). Slices recovered in carbogen buffered high magnesium holding ACSF (ACSF + 1 mM $MgCl_2 \cdot 6H_2O$, 1 mM sodium ascorbate, and 3 mM sodium pyruvate) at room temperature for 1 hr. For imaging, the slices were transferred into an imaging chamber and perfused with 35°C warm ACSF (125 mM NaCl, 3 mM KCl, 1.25 mM $NaH_2PO_4 \cdot H_2O$, 1 mM $MgCl_2 \cdot 6H_2O$, 2 mM $CaCl_2$, 25 mM $NaHCO_3$, and 25 mM glucose) and imaged with an upright laser-scanning confocal microscope (Nikon Eclipse Ni-E) equipped with a 25× water objective (NA 1.1). For green and red fluorescence, the laser lines 488 nm and 561 nm with 525/50 and 595/50 filter sets were used. For stimulation, a concentric bipolar microelectrode was positioned on the beginning of the mossy-fibers in the DG. Datasets were acquired at 1.96 Hz with a pixel size of 0.18 µm at depth of approximately 20–50 µm.

## Primary neuronal cultures

Hippocampi were extracted from E18 WT embryos. After removal of the meninges, hippocampi were collected in Hanks buffered Salt Solution (Sigma, cat. No. H9394, St. Louis, MO, USA) with 7 mM HEPES (Invitrogen, cat. No. 15630–056, Waltham, MA, USA). Neurons were incubated in Hanks-HEPES with 0.25% trypsin (Invitrogen, cat. No. T9253) for 20 min at 37°C. Neurons were washed and triturated with fire polished Pasteur pipettes, then counted in a Fuchs-Rosenthal chamber. Neurons were plated in Neurobasal medium supplemented with 2% B-27 (Invitrogen, cat. No. 11530536), 1.8% HEPES, 0.25% Glutamax (Invitrogen, cat. No. 11574466), and 0.1% Pen/Strep (Invitrogen, cat. No. 11548876). To obtain single neuron cultures, hippocampal neurons were plated in 12-well plates at a density of 1500 cells/well on 18 mm glass coverslips containing micro-islands of rat glia. Micro-islands were generated as described previously (*Meijer et al., 2012*) by plating 8000/well rat glia on UV-sterilized agarose (Type II-A; Sigma, A9918) coated etched glass coverslips stamped with a mixture of 0.1 mg/ml poly-D-lysine (Sigma, P6407), 0.7 mg/ml rat tail collagen (BD Biosciences, 354236, Franklin Lakes, New Jersey), and 10 mM acetic acid (Sigma, 45731).

## Immunocytochemistry for neuronal cultures

Cultures were fixed after 14 days in vitro using 3.7% formaldehyde (Electron Microscopies Sciences, 15680, Hatfield, Pennsylvania) and washed three times with PBS pH 7. Cells were permeabilized for 5 min with 0.5% Triton-X-100 (Fisher Chemical - Thermo Fisher, T/3751/08, Waltham, MA, USA), followed by 30 min incubation with 2% normal goat serum (NGS; Gibco - Thermo Fisher, 16210–072) and 0.1% Triton-X-100 to block aspecific binding. All antibodies were diluted in 2% NGS. The following antibodies were used: chicken anti-MAP2 (1:500, Abcam ab5392, Cambridge, United Kingdom), rabbit anti-Chromogranin B (1:500, SySy 259103, Göttingen, Germany), and rabbit anti-Chromogranin A (1:500, SySy 259003). After three washes with PBS, neurons were incubated with secondary antibodies (1:500; Invitrogen) for 1 hr. After three washes with PBS, coverslips were mounted on microscopic slides with Mowiol-DABCO (Sigma, 81381). Coverslips were imaged using a confocal A1R microscope (Nikon) with LU4A laser unit using a 40× oil immersion objective (NA = 1.3). Images were acquired at 1024×1024 pixels as z-stacks (five steps of 0.25 µm). Maximum projection images were used for analysis. Confocal settings were kept constant for all images within the experiment.

## Processing and analysis of imaging data
### In vivo and slice data
The acquired time-lapse datasets were registered for slight image movement in the $x$–$y$ plane between consecutive images with the Fiji (RRID:SCR 002285) Plugin 'Descriptor based series registration'

(*Preibisch et al., 2010*). After registration, the datasets were background subtracted and median filtered. Axons with moving puncta were identified by making a SD z-projection of the stack which highlights moving puncta as lines. Moving puncta were analyzed using kymographs. Lines fitted over tracks in the kymographs were analyzed with an ImageJ macro written by Fabrice P. Cordelières (*Zala et al., 2013*). For the analysis of axons infected with two markers, only axon stretches with at least three moving puncta and 1 per direction of each marker were analyzed. Relative flux was calculated as percentage of the main vector of movement of all tracks.

### In vitro data

For co-localization analysis, morphological masks were drawn using SynD (*Schmitz et al., 2011*) and imported in ImageJ to remove background fluorescence. Co-localization between pHluorin signal and ChgA/B was measured using the JACoP plugin in ImageJ. Thresholds were set manually per cell.

## Statistics

Basic statistics were calculated using Prism (GraphPad, RRID:SCR 002798). Most analyzed data did not show a Gaussian distribution. Hence, distributions of the datasets were compared with non-parametric two-sided Kolmogorov-Smirnov tests and Wilcoxon signed-rank test. A p-value below 0.05 was considered statistically significant. To determine the influence of the location (synapse/shaft) and activity state on to organelle trafficking, a linear mixed effect model was applied using R (RRID:SCR_001905; RStudio, RRID:SCR_000432). The influence of mouse number, dataset number, and track number was included as random effects, while the fit of the models with and without location/activity state as fixed effect was compared to each other using ANOVA. A p-value below 0.05 was considered statistically significant.

## Acknowledgements

We thank Joke Wortel and Robbert Zalm for the extensive help with animal experiments and production of viral particles. We thank Aygul Subkhangulova for testing the signal-dead NPY reporter and Jessie Brunner for advice on statistical analyses. This work was supported by a European Research Council (ERC) Advanced grant (322966) of the European Union (to MV). COSYN (Comorbidity and Synapse Biology in Clinically Overlapping Psychiatric Disorders); the NWO Gravitation program BRAINSCAPE: A Roadmap from Neurogenetic to Neurobiology (NWO: 024.004.012, to MV) and the Netherlands Scientific Organisation and De Hersenstichting (013-17-002), under the frame of the Neuron Cofund ERA-Net SNAREopathy (to RFT).

## Additional information

### Funding

| Funder | Grant reference number | Author |
| --- | --- | --- |
| European Research Council | Advanced grant (322966) | Matthijs Verhage |
| Nederlandse Organisatie voor Wetenschappelijk Onderzoek | Gravitation program BRAINSCAPES (NWO: 024.004.012) | Matthijs Verhage |
| Nederlandse Organisatie voor Wetenschappelijk Onderzoek | 013-17-002 Neuron Cofund ERA-Net SNAREopathy | Ruud F Toonen |
| Hersenstichting | 013-17-002 Neuron Cofund ERA-Net SNAREopathy | Ruud F Toonen |

The funders had no role in study design, data collection and interpretation, or the decision to submit the work for publication.

## Author contributions
Joris P Nassal, Conceptualization, Data curation, Formal analysis, Investigation, Visualization, Methodology, Writing – original draft, Writing – review and editing; Fiona H Murphy, Data curation, Formal analysis, Investigation, Visualization, Methodology, Writing – review and editing; Ruud F Toonen, Conceptualization, Supervision, Project administration, Writing – review and editing; Matthijs Verhage, Conceptualization, Supervision, Funding acquisition, Writing – original draft, Project administration, Writing – review and editing

## Author ORCIDs
Fiona H Murphy ⓘ http://orcid.org/0000-0002-3995-0607
Ruud F Toonen ⓘ http://orcid.org/0000-0002-9900-4233
Matthijs Verhage ⓘ http://orcid.org/0000-0002-6085-7503

## Ethics
All animal experiments were approved by the national committee for animal research in the Netherlands (Centrale Commissie Dierenproven; AVD112002017824), the local animal research committee of the VU University Amsterdam (Dierexperimentencommissie) and the animal welfare body of the VU/VUmc Amsterdam (Instantie voor Dierenwelzijn) and were carried out in accordance with the European Communities Council Directive (010/63/EU).

## Decision letter and Author response
Decision letter https://doi.org/10.7554/eLife.81721.sa1
Author response https://doi.org/10.7554/eLife.81721.sa2

---

# Additional files

## Supplementary files
• MDAR checklist

## Data availability
Analysis files as well as original imaging datasets have been made publicly available via the DataverseNL project (https://dataverse.nl) under the DOI: https://doi.org/10.34894/9QYXZS.

The following dataset was generated:

| Author(s) | Year | Dataset title | Dataset URL | Database and Identifier |
|---|---|---|---|---|
| Nassal JP, Murphy FH, Toonen RF, Verhage M | 2022 | Replication Data for: Differential axonal trafficking of Neuropeptide Y-, LAMP1- and RAB7-tagged organelles in vivo | https://doi.org/10.34894/9QYXZS | DataverseNL, 10.34894/9QYXZS |

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
