## [Editor Report]

This is an important, well-written and easily comprehended quantitative imaging study that analyzes the motion of endo-lysosomal compartments within axons in vivo using simultaneous multiphoton imaging in the mammalian brain. The simultaneous dual two-photon imaging is well-executed and represents a substantive advance in a field that relies heavily on in vitro neuronal culture preparations. The authors address an issue of cell polarity, providing strong support for their ability to determine directional movement (anterograde versus retrograde), and characterize interesting differences in motion, including activity-dependent and calcium-dependent alterations at or near synapses.

---

## [Decision Letter]

**Decision letter after peer review:**

Thank you for submitting your article "Differential axonal trafficking of Neuropeptide Y-, LAMP1- and RAB7-tagged organelles in vivo" for consideration by *eLife*. Your article has been reviewed by 3 peer reviewers, one of whom is a member of our Board of Reviewing Editors, and the evaluation has been overseen by a Reviewing Editor and Suzanne Pfeffer as the Senior Editor. The following individual involved in the review of your submission has agreed to reveal their identity: Thierry Galli (Reviewer #2).

During a discussion of the manuscript, it has become apparent that the manuscript could be substantially strengthened by placing the new work more clearly in the context of what has been done previously, particularly in other central and peripheral nervous system preparations in other model organisms. Beyond that, the comments of reviewer #3 could primarily be addressed through text clarification, additional data analysis, or the inclusion of additional data that may already have been acquired, or which might be straightforward to acquire in a short time frame. Very few of these comments and criticisms specifically request the addition of new data. As is readily apparent, reviewers #1 and #2 were generally quite positive about the study. Addressing the questions of reviewer #3 could elevate the work even further. It was generally agreed upon that no new "mechanistic" data are being requested. A focus should remain on context, clarifications, and additional analyses when possible.

*Reviewer #1 (Recommendations for the authors):*

My only criticism is that the simultaneous multi-photon approach is not applied in a manner that is absolutely crucial for the final quantifications.

*Reviewer #2 (Recommendations for the authors):*

In order to establish this article as a reference for future studies in the most straightforward way, this reviewer would like to suggest adding a table with the velocities indicated in the discussion and a comparison with previous studies carried out in cultured neurons.

Experiments in mutant mice and more markers would definitively strengthen this article but this reviewer also understands the delay that such a request would induce.

*Reviewer #3 (Recommendations for the authors):*

General comments:

(1) As outlined in sections 1 and 2 of the public review, I feel this work mostly is incremental and corroborative in nature. In their previous collaborative work with the Kuner lab (Knabbe et al. J Physiol 2018) the authors have already described the in vivo trafficking of DCVs, using very similar experiments to show a predominance of anterograde transport and a delay at presynaptic sites. Thus, in this work, novelty is mostly related to the comparison of DCVs with late endosomes and lysosomes – but this part is relatively limited and weak. First, the markers used are not undisputed (see recent Lie et al. paper by the Nixon lab that the authors cite, but do not really discuss) and not further characterized here. The experiment of differentially tagging the two markers and actually measuring overlap is omitted – so I cannot decide, whether DCVs are compared here to one or two other organelle classes (if that descriptor even holds for the LE/L continuum). The results on the plus tip trackers and the presynaptic labels are as expected, and again overlap with prior work – the possibility of corroborating the synapse finding with a LE/L marker was not pursued. Similarly, the calcium imaging experiments were only performed with DCVs, and not with LE/Ls. This – and the low temporal resolution, which makes a detailed trafficking analysis difficult – does not allow concluding, whether the observed effects are cargo-specific (and therefore more likely to be due to organelle or motor characteristics), as opposed to general effects (that could be ascribed more to cytoskeletal effects or organelle crowding given the similar sizes of the two cargoes). A detailed correlation analysis of trafficking of two cargo classes across the same axon stretch and the same activity patterns could have been quite informative, in my view, but would not necessarily lift the work to general interest.

(2) Regarding the effects of activity, which are potentially interesting. One would have to be really sure that all the statistical corrections needed were applied for the multiple conditions; in several instances, there are effects in similar directions and sizes across time points, but these fail to reach the significance criterion, suggesting to me that the study might in part be underpowered to fully reveal what is going on – but in any case, the effect size is probably not drastic. There is an additional caveat here: As the DVC is an organelle that is 'consumed' in an activity-dependent manner, the 'effects' of activity could be entirely on the level of which organelles make it to the site of observation, rather than any local effect. So for example, slower moving organelles could be more likely to be captured at more proximal synapses, and this could create these weak effects – the complex patterns in which DCVs could potentially traffic through axons have been well described in flies. Observations on LE/Ls could have been illuminating here, as this caveat of activity-dependent consumption perhaps would not apply in the same way and effects that are organelle-specific could have been revealed.

(3) The fact that 'resting' organelles could not be included in the analysis (P13, L5: "Only moving fluorescent puncta were analyzed to prevent the inclusion of released fluorophore, organelles from other axons, and auto-fluorescent background") makes an interpretation of the overall motility difficult. It has been pointed out (including by the authors) that for mitochondria, only a small fraction actually moves in mature axons in vivo (they cite this as a motivation for in vivo studies of organelle dynamics). The data do not really allow a judgement, whether this is also true for the organelles investigated here, as the size of the non-moving population of organelles is not determined. I also wonder how representative the measured trafficking behavior is, given that in most experiments, no separate structural marker was used to identify axons. I assume the organelle density and trafficking might have played a role in selecting included axon stretches, which might potentially have biased the sites of observation.

Specific comments

(4) Related to Point 1 above: I do not see a firm experimental basis for the following statements – P27, L10: "RAB7 and LAMP1 might partially overlap in the organelles they label. However, the different trafficking properties observed here suggest these markers label different subpopulations of organelles."; P28, L7 "The three classes of organelles showed substantial differences in trafficking speeds." – but: Figure 8

(5) P1, L14: "Different organelles traveling through neurons exhibit distinct properties in vitro, but this has not been investigated in the intact brain." This suggests that the trafficking of different organelles has not been looked at in vivo – while one can probably argue, whether 'in the brain' is the key qualifier here if one sticks with CNS, there have been studies that have tracked different organelles (e.g. in the mouse spinal cord). Also, the discussion perhaps undervalues the amount of in vivo or ex vivo work done in zebrafish and invertebrates. These studies mostly did not have the explicit purpose to compare movement behaviors, but the fact that such behavior is highly distinct for different organelles can be deduced from the existing literature. So perhaps it would be fair to tone down the novelty claim a bit here.

(6) P2, L25: "Lysosomal proteins are delivered into the axon by endosomal organelles and fuse with autophagosomes to form lysosomes." This sentence suggests that a fusion event with an autophagosome is a necessary step in lysosome biogenesis; this does not match with how I understand this process. The authors might want to check this statement, and in this process also position themselves a bit more clearly regarding the ongoing debate on the presence of lysosomes in axons.

(7) P3, L11: "We also show that transport of DCVs slows down in and close to synapses. Furthermore, we show that increased calcium levels lead to a delayed increase of the transport speed of NPY-tagged organelles." – What does "delayed increase" mean here?

(8) 16, L9: "D: Mean speed of Lamp1-tagged organelles in MacF18 co-expressing axons (8 different axons, 78 tracks)." – Relabel axis and indicate as paired analysis to support P15, L12 "In every axon the direction of the average faster-trafficking Lamp1 puncta was the same as the anterograde (towards plus-end) direction of the MacF18-GFP puncta (Figure 3B)."

(9) P21, L23: Why is the overexpression of an active NPY not a general problem also for the comparison with LAMP1 and Rab7, but only in the experiments where the NPYsd variant was used? P21, L23: "The NPYsd variant was used to ensure that the released NPY had no effect on local circuits."

(10) P21, L28: What does the "active state" represent in electrophysiological terms? The Ca signals are not very sharp (Figure 6B). P21, L28: "An active state of an axoGCaMP-expressing axon was defined as time in which the axoGCaMP signal was above half-maximal DF/F (Figure 6B)." The activity effects on trafficking do not seem very clear to me. They could be due to many changes unrelated to the discussed signalling affecting the transport machinery itself (see above point 2, but also general cytoskeletal changes), but are probably also confounded by the vague definition of 'active state'. Perhaps there is a more convincing dose-response if one tried to more precisely measure and define activity, at least in the slice preparation (which I otherwise do not find very illuminating in the context of a paper that stresses the need for work in an intact brain)?

---

## [Author Response]

Reviewer #1 (Recommendations for the authors):My only criticism is that the simultaneous multi-photon approach is not applied in a manner that is absolutely crucial for the final quantifications.

We thank reviewer 1 for the time dedicated to our manuscript. We are very thankful for the positive comments regarding our work and agree that the approach used has the potential to benefit the field in transferring in vitro insight to in vivo cell biology. The multi-photon approach in our mind is necessary to provide real comparability between different organelles within a single axon. In these in vivo experiments, we inevitably have a long distance between the lens and the axon, and, in addition, there is a lot of out-of-focus fluorescence. For these two reasons we think the multi-photon approach is the best option.

Reviewer #2 (Recommendations for the authors):In order to establish this article as a reference for future studies in the most straightforward way, this reviewer would like to suggest adding a table with the velocities indicated in the discussion and a comparison with previous studies carried out in cultured neurons.Experiments in mutant mice and more markers would definitively strengthen this article but this reviewer also understands the delay that such a request would induce.

We thank reviewer 2 for the time dedicated to our manuscript. We are very thankful for the complements regarding our work and definitely agree that future experiments should include many more markers. We also agree that this type of approach would be great to study mouse mutants specifically mouse lines with trafficking deficits.

We agree with the suggestion of reviewer 2 and we added a supplemental table including our velocities and comparing them with multiple studies in different model systems (new Table 1).

Reviewer #3 (Recommendations for the authors):General comments:(1) As outlined in sections 1 and 2 of the public review, I feel this work mostly is incremental and corroborative in nature. In their previous collaborative work with the Kuner lab (Knabbe et al. J Physiol 2018) the authors have already described the in vivo trafficking of DCVs, using very similar experiments to show a predominance of anterograde transport and a delay at presynaptic sites. Thus, in this work, novelty is mostly related to the comparison of DCVs with late endosomes and lysosomes – but this part is relatively limited and weak. First, the markers used are not undisputed (see recent Lie et al. paper by the Nixon lab that the authors cite, but do not really discuss) and not further characterized here. The experiment of differentially tagging the two markers and actually measuring overlap is omitted – so I cannot decide, whether DCVs are compared here to one or two other organelle classes (if that descriptor even holds for the LE/L continuum). The results on the plus tip trackers and the presynaptic labels are as expected, and again overlap with prior work – the possibility of corroborating the synapse finding with a LE/L marker was not pursued. Similarly, the calcium imaging experiments were only performed with DCVs, and not with LE/Ls. This – and the low temporal resolution, which makes a detailed trafficking analysis difficult – does not allow concluding, whether the observed effects are cargo-specific (and therefore more likely to be due to organelle or motor characteristics), as opposed to general effects (that could be ascribed more to cytoskeletal effects or organelle crowding given the similar sizes of the two cargoes). A detailed correlation analysis of trafficking of two cargo classes across the same axon stretch and the same activity patterns could have been quite informative, in my view, but would not necessarily lift the work to general interest.

We believe that our work provides substantially new insights compared to our previous work. We not only imaged for the first time two organelle markers, but also two functional markers and corroborated findings in invertebrate model systems.

We agree that “the markers used are not undisputed”. Exactly for this reason, we have not interpreted our marker imaging data and we refer throughout the manuscript to NPY-tagged organelles, Rab7-tagged organelles etc. We now address the dispute about the LE/L markers regarding the Lie et al. paper in more detail in the discussion (p 28, line 5).

We are not so sure that higher temporal resolution would yield new insights. None of our conclusions requires higher sampling rates. Furthermore, by observing trafficking characteristics in each channel, quantitatively comparing these and performing the appropriate statistical tests, we are convinced we can conclude “whether the observed effects are cargo-specific [or not]”.

(2) Regarding the effects of activity, which are potentially interesting. One would have to be really sure that all the statistical corrections needed were applied for the multiple conditions; in several instances, there are effects in similar directions and sizes across time points, but these fail to reach the significance criterion, suggesting to me that the study might in part be underpowered to fully reveal what is going on – but in any case, the effect size is probably not drastic. There is an additional caveat here: As the DVC is an organelle that is 'consumed' in an activity-dependent manner, the 'effects' of activity could be entirely on the level of which organelles make it to the site of observation, rather than any local effect. So for example, slower moving organelles could be more likely to be captured at more proximal synapses, and this could create these weak effects – the complex patterns in which DCVs could potentially traffic through axons have been well described in flies. Observations on LE/Ls could have been illuminating here, as this caveat of activity-dependent consumption perhaps would not apply in the same way and effects that are organelle-specific could have been revealed.

We believe that by choosing a mixed linear effect model analysis, we have adequately included other possible factors for variation and we “all the statistical corrections […] were applied for the multiple conditions [to make solid conclusions]”.

With regard to the activity effect on DCV possibly being explained by subpopulations of vesicles being captured in proximal synaptic sites: We randomly imaged axon stretches and therefore studied different axon parts. Furthermore, we control for speed/activity within the same axon stretches (speed during activity and during rest) and therefore exclude this hypothesis.

(3) The fact that 'resting' organelles could not be included in the analysis (P13, L5: "Only moving fluorescent puncta were analyzed to prevent the inclusion of released fluorophore, organelles from other axons, and auto-fluorescent background") makes an interpretation of the overall motility difficult. It has been pointed out (including by the authors) that for mitochondria, only a small fraction actually moves in mature axons in vivo (they cite this as a motivation for in vivo studies of organelle dynamics). The data do not really allow a judgement, whether this is also true for the organelles investigated here, as the size of the non-moving population of organelles is not determined. I also wonder how representative the measured trafficking behavior is, given that in most experiments, no separate structural marker was used to identify axons. I assume the organelle density and trafficking might have played a role in selecting included axon stretches, which might potentially have biased the sites of observation.

We agree with the reviewer that an analysis of the non-moving DCV fraction (stable for multiple minutes) would be informative. However, several crucial arguments, also cited by the reviewer, preclude this.

With regard to the measured trafficking behaviour being representative: We randomly choose axon stretches in the datasets purely with the prerequisite that they contained multiple moving organelles of both markers. There are large variations in organelle density and trafficking characteristics between single analysed axon. We believe to have sampled data in an unbiased manner.

Specific comments(4) Related to Point 1 above: I do not see a firm experimental basis for the following statements – P27, L10: "RAB7 and LAMP1 might partially overlap in the organelles they label. However, the different trafficking properties observed here suggest these markers label different subpopulations of organelles."; P28, L7 "The three classes of organelles showed substantial differences in trafficking speeds." – but: Figure 8

We agree that these statements are probably too strong and adjusted them (p 28, line 13; p 30, line 1).

(5) P1, L14: "Different organelles traveling through neurons exhibit distinct properties in vitro, but this has not been investigated in the intact brain." This suggests that the trafficking of different organelles has not been looked at in vivo – while one can probably argue, whether 'in the brain' is the key qualifier here if one sticks with CNS, there have been studies that have tracked different organelles (e.g. in the mouse spinal cord). Also, the discussion perhaps undervalues the amount of in vivo or ex vivo work done in zebrafish and invertebrates. These studies mostly did not have the explicit purpose to compare movement behaviors, but the fact that such behavior is highly distinct for different organelles can be deduced from the existing literature. So perhaps it would be fair to tone down the novelty claim a bit here.

We agree and have specified the statement to “intact mammalian brain” (p 1, line 15).

(6) P2, L25: "Lysosomal proteins are delivered into the axon by endosomal organelles and fuse with autophagosomes to form lysosomes." This sentence suggests that a fusion event with an autophagosome is a necessary step in lysosome biogenesis; this does not match with how I understand this process. The authors might want to check this statement, and in this process also position themselves a bit more clearly regarding the ongoing debate on the presence of lysosomes in axons.

We agree that the sentence might be misleading. This process is only necessary to degrade autophagosome cargo within autolysosomes (Cheng XT, Zhou B, Lin MY, Cai Q, Sheng ZH. Axonal autophagosomes recruit dynein for retrograde transport through fusion with late endosomes. J Cell Biol. 2015 May 11;209(3):377-86. doi: 10.1083/jcb.201412046. Epub 2015 May 4). We therefore changed the sentence (p 2, line 24). We cannot add new evidence to the debate about degradative lysosomes in axons and therefore refrain from commenting on it. However, we emphasize that we observe classical lysosome markers in organelle trafficking in thalamo-cortical axons in vivo.

(7) P3, L11: "We also show that transport of DCVs slows down in and close to synapses. Furthermore, we show that increased calcium levels lead to a delayed increase of the transport speed of NPY-tagged organelles." – What does "delayed increase" mean here?

"delayed increase" was meant to indicate delayed in time. We agree that is not directly obvious and removed `delayed´, which is not necessary in the short summary. We have adjusted this statement (p 3, line 12).

(8) 16, L9: "D: Mean speed of Lamp1-tagged organelles in MacF18 co-expressing axons (8 different axons, 78 tracks)." – Relabel axis and indicate as paired analysis to support P15, L12 "In every axon the direction of the average faster-trafficking Lamp1 puncta was the same as the anterograde (towards plus-end) direction of the MacF18-GFP puncta (Figure 3B)."

We added a panel C to figure 3 showing a paired analysis per axon.

(9) P21, L23: Why is the overexpression of an active NPY not a general problem also for the comparison with LAMP1 and Rab7, but only in the experiments where the NPYsd variant was used? P21, L23: "The NPYsd variant was used to ensure that the released NPY had no effect on local circuits."

We agree that the overexpression of normal NPY could theoretically affect the co-trafficking data. We choose this construct because we wanted comparability to previous trafficking studies using the same overexpression constructs. The possible change of activity state would affect both labelled organelles and therefore still allow these comparisons. We switched to the signal-dead construct, when specifically assessing effects on trafficking. We hypothesized these to be due to activity changes and therefore specifically wanted to exclude possible effects of release overexpressed NPY.

(10) P21, L28: What does the "active state" represent in electrophysiological terms? The Ca signals are not very sharp (Figure 6B). P21, L28: "An active state of an axoGCaMP-expressing axon was defined as time in which the axoGCaMP signal was above half-maximal DF/F (Figure 6B)." The activity effects on trafficking do not seem very clear to me. They could be due to many changes unrelated to the discussed signalling affecting the transport machinery itself (see above point 2, but also general cytoskeletal changes), but are probably also confounded by the vague definition of 'active state'. Perhaps there is a more convincing dose-response if one tried to more precisely measure and define activity, at least in the slice preparation (which I otherwise do not find very illuminating in the context of a paper that stresses the need for work in an intact brain)?

We agree that it would be great to study more precise activity traces and that this might lead to a better understanding of the effects we found. However, to be able to image these small organelles the imaging frequency has a limit. The small amount of fluorophore per organelle, the desired imaging area size and the desired total imaging time led to those limits. We therefore were limited to a comparatively slow imaging frequency for calcium imaging. This led us to the decision for a binary activity threshold which is quite often used in the field. The decision to do a similar analysis in slices was made to be able to easily compare those results with the in vivo results.